# Associations between introduction and withdrawal of a financial incentive and timing of attendance for antenatal care and incidence of small for gestational age: natural experimental evaluation using interrupted time series methods

Jean Adams,[1] Zelda van der Waal,[2] Steven Rushton,[2] Judith Rankin[3]

[1]MRC Epidemiology Unit, Centre for Diet and Activity Research, University of Cambridge, Cambridge, UK
[2]School of Biology, Newcastle University, Newcastle upon Tyne, UK
[3]Institute of Health and Society, Newcastle University, Newcastle upon Tyne, UK

**Correspondence to**
Dr Jean Adams;
jma79@medschl.cam.ac.uk

## ABSTRACT

**Objectives** To determine whether introduction or withdrawal of a maternal financial incentive was associated with changes in timing of first attendance for antenatal care ('booking'), or incidence of small for gestational age.

**Design** A natural experimental evaluation using interrupted time series analysis.

**Setting** A hospital-based maternity unit in the north of England.

**Participants** 34 589 women (and their live-born babies) who delivered at the study hospital and completed the 25th week of pregnancy in the 75 months before (January 2003 to March 2009), 21 months during (April 2009 to December 2010) and 36 months after (January 2011 to December 2013) the incentive was available.

**Intervention** The Health in Pregnancy Grant was a financial incentive of £190 ($235; €211) payable to pregnant women in the UK from the 25th week of pregnancy, contingent on them receiving routine antenatal care.

**Primary and secondary outcome measures** The primary outcome was mean gestational age at booking. Secondary outcomes were proportion of women booking by 10, 18 and 25 weeks' gestation; and proportion of babies that were small for gestational age.

**Results** By 21 months after introduction of the grant (ie, immediately prior to withdrawal), compared with what was predicted given prior trends, there was an reduction in mean gestational age at booking of 4.8 days (95% CI 2.3 to 8.2). The comparable figure for 24 months after withdrawal was an increase of 14.0 days (95% CI 2.8 to 16.8). No changes in incidence of small for gestational age babies were seen.

**Conclusions** The introduction of a universal financial incentive for timely attendance at antenatal care was associated with a reduction in mean gestational age at first attendance, but not the proportion of babies that were small for gestational age. Future research should explore the effects of incentives offered at different times in pregnancy and of differing values; and how stakeholders view such incentives.

### Strengths and limitations of this study

► We used interrupted time series methods to evaluate this natural experiment; one of the strongest quasiexperimental research designs available.

► By including substantial data before and after interventions, we took account of underlying secular trends.

► However, interrupted time series designs are observational and we cannot categorically ascribe the changes documented to the intervention.

► One of our secondary outcomes was proportion of babies born small for gestational age—a substantial improvement on previous studies that use a simple low birth weight cut-off.

► Differences between women included and excluded from the analyses may limit external validity, as may our use of data from only one hospital.

## INTRODUCTION

Financial incentives are increasingly used to encourage health-promoting behaviours. However, few large, pragmatic evaluations in high-income countries have been conducted.[1 2]

The Health in Pregnancy Grant (HiPG) was introduced in April 2009 as a one-off payment of £190 ($235; €211) payable to all pregnant women, normally resident in the UK, after the 25th week of pregnancy, but before delivery. Women submitted a claim form, signed by their doctor or midwife confirming their expected delivery date and that they had received usual antenatal care.[3] A key aim of the HiPG was to act as an 'incentive to seek the recommended health advice at the appropriate time'.[3] Following a general election in 2010, the HiPG was withdrawn

**Table 1** Characteristics of those included and excluded from the analytical cohort

| Variable | Level | Before HiPG availability (January 2003 to March 2009) | | During HiPG availability (April 2009 to December 2010) | | After HiPG availability (January 2011 to December 2013) | | Full study period (January 2003 to December 2013) | |
|---|---|---|---|---|---|---|---|---|---|
| | | Included* | Excluded† | Included* | Excluded† | Included* | Excluded† | Included* | Excluded† |
| n (%) | | 18 744 | 2816 | 6126 | 862 | 9719 | 1304 | 34 589 | 4982 |
| Maternal age, n (%) | <25 years | 6359 (33.9) | 1327 (47.1) | 2039 (33.3) | 374 (43.4) | 2836 (29.2) | 523 (40.1) | 11 234 (32.5) | 2224 (44.6) |
| | 25–34 years | 9684 (51.7) | 1208 (42.9) | 3272 (53.4) | 405 (47.0) | 5577 (57.4) | 651 (49.9) | 18 533 (53.6) | 2264 (45.4) |
| | 35+ years | 2701 (14.4) | 281 (10.0) | 815 (13.3) | 83 (9.6) | 1306 (13.4) | 130 (10.0) | 4822 (13.9) | 494 (9.9) |
| | Data not available, n (%) | 0 | 0‡ | 0 | 0‡ | 0 | 0‡ | 0 | 0‡ |
| Parity, n (%) | 0 | 8077 (43.1) | 1288 (45.7) | 2670 (43.6) | 378 (43.9) | 3944 (40.6) | 585 (44.9) | 14 691 (42.5) | 2251 (45.2) |
| | 1+ | 10 667 (56.9) | 1528 (54.3) | 3456 (56.4) | 484 (56.2) | 5775 (59.4) | 719 (55.1) | 19 898 (57.5) | 2731 (54.8) |
| | Data not available, n (%) | 0 | 0‡ | 0 | 0 | 0 | 0‡ | 0 | 0‡ |
| Index of multiple deprivation group, n (%) | Most deprived | 10 820 (57.7) | 1777 (63.1) | 3566 (58.2) | 547 (63.5) | 5821 (59.9) | 794 (60.9) | 20 207 (58.4) | 3118 (62.6) |
| | Moderately deprived | 4213 (22.5) | 490 (17.4) | 1330 (21.7) | 160 (18.6) | 2230 (22.9) | 220 (16.9) | 7773 (22.5) | 870 (17.5) |
| | Least deprived | 3711 (19.8) | 361 (12.8) | 1230 (20.1) | 92 (10.7) | 1668 (17.2) | 128 (9.8) | 6609 (19.1) | 581 (11.7) |
| | Data not available, n (%) | 0 | 188 (6.7)‡ | 0 | 63 (7.3)‡ | 0 | 162 (12.4)‡ | 0 | 413 (8.3)‡ |
| Study outcomes | Mean (SD) gestational age at booking, days | 76.7 (35.8) | 106.2 (45.5)§ | 77.8 (41.3) | 99.3 (39.8)§ | 71.8 (31.8) | 95.0 (39.1)§ | 75.6 (35.8) | 102.0 (43.2)§ |
| | Booked by 10 weeks, n (%) | 10 261 (54.7) | 442 (17.0)‡ | 3540 (57.8) | 152 (18.8)‡ | 6127 (63.0) | 283 (23.9)‡ | 19 928 (57.6) | 877 (19.1)‡ |
| | Booked by 18 weeks, n (%) | 17 579 (93.8) | 2012 (77.7)‡ | 5675 (92.6) | 676 (83.7)‡ | 9307 (95.8) | 1022 (86.3)‡ | 32 561 (94.1) | 3719 (81.0)‡ |
| | Booked by 25 weeks, n (%) | 18 098 (96.6) | 2397 (92.1)‡ | 5846 (95.4) | 766 (94.8) | 9477 (97.5) | 1129 (95.4)‡ | 33 421 (96.6) | 4292 (93.4)‡=3 |
| | Small for gestational age, n (%) | 2346 (12.5) | 17 (8.3) | 743 (12.1) | 7 (11.5) | 1163 (12.0) | 25 (14.9) | 4252 (12.3) | 49 (11.3) |
| | Data not available on time at booking, n (%) | 0 | 214 (7.6) | 0 | 54 (6.3) | 0 | 120 (9.2) | 0 | 388 (7.8) |
| | Data not available on birth weight, n (%) | 0 | 2611 (92.7) | 0 | 801 (92.9) | 0 | 1136 (87.1) | 0 | 4548 (91.3) |

*Women who delivered at the study hospital were known to have reached the 25th week of gestation between 1 January 2003 and 31 December 2013 and had available data on all variables of interest.

†Women who delivered at the study hospital were known to have reached the 25th week of gestation between 1 January 2003 and 31 December 2013 and did not have available data on all variables of interest.

‡X² test indicates difference in distribution of levels between included and excluded at a level of P<0.05.

§T-test indicates difference in means between included and excluded at a level of P<0.05.

HiPG, Health in Pregnancy Grant.

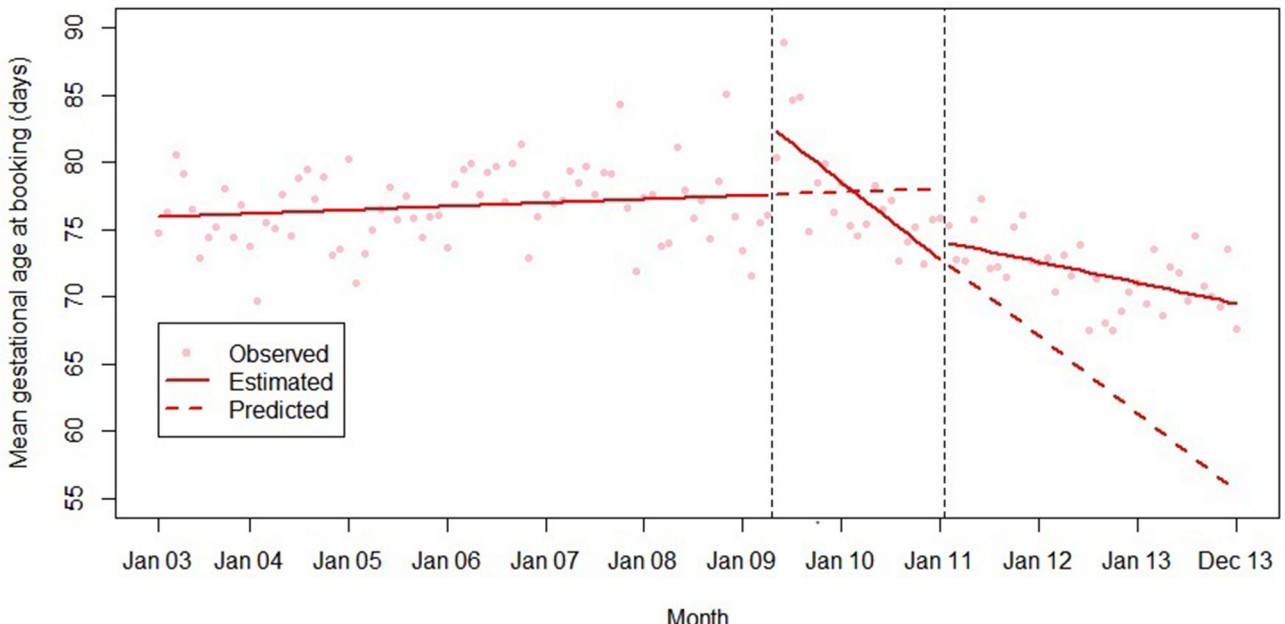

**Figure 1** Summary of interrupted time series model of the introduction and withdrawal of the Health in Pregnancy Grant on mean gestational age at booking (days).

with women only able to claim if they reached the end of the 25th week of pregnancy before 1 January 2011.

England compares poorly with other European countries on perinatal outcomes.[4] One possible reason is poor attendance at antenatal care, which is associated with increased risk of small for gestational age (SGA),[5 6] and a range of adverse outcomes.[7–9] National guidance recommends that the first antenatal (or 'booking') visit should ideally take place by 10 weeks' gestation and, at the latest, by 18 weeks.[10] Women living in more deprived circumstances tend to book later in pregnancy.[11]

Health-promoting financial incentives may be more effective in promoting one-off behaviours than complex behaviour change.[12–15] Antenatal care is a series of one-off behaviours and may be particularly responsive to incentives. However, a recent systematic review found only five trials of maternal incentives for antenatal care—three conducted in the USA and one each in Mexico and Honduras.[16] No effect on timing of antenatal care was found (although only one study investigated this).[17] No studies included birth weight or SGA as outcomes. A further observational study from the USA found no effect of an incentive on incidence of low birth weight.[18] One recent evaluation of the HiPG in Scotland reported no effect on birth weight, but a positive effect on the proportion of women booking by 25 weeks (other aspects of timing of attendance were not studied).[19]

UK public health policymakers and members of the public think that it may be appropriate to target financial incentives at people living in more deprived circumstances—perhaps because those living in more deprived circumstances are more in need of financial support.[20 21] There is some systematic review evidence that people living in more deprived circumstances may

be more responsive to fiscal interventions in general. Other personal characteristics, such as age and previous experience of the behaviour incentivised, may also influence responsiveness. However, differential responses to health-promoting financial incentives between population groups have not been systematically studied.[1]

The introduction and withdrawal of the HiPG provided a unique opportunity for a large-scale, pragmatic, natural experimental evaluation of a health-promoting financial incentive.[22] Our research questions were: was the introduction or withdrawal of the HiPG associated with a change in the timing of booking, or incidence of SGA? Did any effect of the HiPG vary according to maternal age, parity or socio-economic position?

## METHODS
We used an interrupted time series (ITS) design.

### Data and inclusion criteria
We used routine data from a maternity unit in a tertiary hospital in northern England, extracted in May 2015. The study hospital is a general teaching hospital with over 1000 beds in a town with a population of ~175 000 people. Both the town and the surrounding areas are more deprived than the English average.

Participants were women (and their live-born babies) who delivered at the study hospital and were known to have completed the 25th week of pregnancy in the 75 months before (January 2003 to March 2009) introduction of the HiPG, 21 months during (April 2009 to December 2010) availability of the HiPG and 36 months after (January 2011 to December 2013) withdrawal of the HiPG. The time periods included were pragmatically arrived at based

on when data were available from, and when the HiPG was introduced and withdrawn. Our final data set of 132 monthly data points (and a mean of 262 cases per data point—see the Results section) substantially exceeds the minimum requirements for ITS (of at least eight data points per intervention phase and 100 individual observations per data point).[23] Aggregating to the weekly, rather than monthly, level would not have achieved these requirements—with a mean of 60 cases in each of 572 weekly data points. As calculation of when women reached the 25th week of pregnancy depended on knowing the date of their last menstrual period (LMP), women for whom this date was missing were excluded.

Women who had a termination or experienced a stillbirth were excluded, as were women with missing data on any variable of interest. Women who delivered more than one live baby in any one pregnancy, or had more than one pregnancy that resulted in a live birth during the study period, were included with each baby counted as a separate 'case'. As we did not have access to any identifiable data on women, we were not able to determine on how many occasions this occurred or to take it into account in modelling.

### Outcome measures

Our outcome measures focus on the stated aim of the HiPG—to encourage women to 'seek the recommended health advice at the appropriate time'.[3] The primary outcome was mean gestational age at booking, calculated from dates of booking (recorded by antenatal care staff) and LMP (self-reported). As national guidance recommends booking ideally before 10 weeks, and definitely before 18 weeks, and the HiPG was available to women from the 25th week, the proportion of women booking by 10, 18 and 25 weeks' gestation were secondary outcomes.[10]

As timely attendance for antenatal care is thought to improve perinatal outcomes, we included a final secondary outcome: proportion of babies that were SGA. It should be noted that there is likely to be a long and complicated chain of causation, if any, between receiving the HiPG and changes in gestational weight for age. We defined SGA as birth weight z-score below the 10th percentile for sex-specific gestational age.[24] This was calculated using infant sex, birth weight and dates of LMP and delivery (all except LMP recorded by antenatal care and delivery staff).

### Other variables of interest

We studied whether any effects of the HiPG on the outcomes varied according to maternal age at delivery (in years, calculated from maternal date of birth and date of delivery and divided into three groups: <25, 25–34, or 35+ years), parity (self-reported and considered as 0 or 1+ in analyses) and socioeconomic position. The main age group (ages 25–34 years) was coded using mid-decade to mid-decade as the convention recommended to increase comparability between studies. We did not further subdivide the other age groups as only eight women in the

included sample were aged less than 15 years and only 27 were aged more than 44 years. Socioeconomic position was measured using the Index of Multiple Deprivation (IMD) 2007 rank assigned to maternal address at delivery.[25] IMD is an area-based measure of deprivation and ranks were divided into thirds for analysis based on the distribution across England.

### Data preparation

Data cleaning aimed to exclude data that were implausible. Date of LMP was recorded as month and year only in around 20% of cases. To include these cases, day of month was set to the 1st. Gestational age at first antenatal care of less than 28 days (4 weeks) or more than 308 days (44 weeks), gestational age at delivery of less than 24 weeks or more than 44 weeks, or birth weight z-scores of less than −3 or more than 3 were recoded as missing as these are likely to represent recording or transcription errors.[24]

### Data analyses

We first compared women in the data set who did and did not meet the inclusion criteria using $\chi^2$ and t-tests.

For the main analysis, an uncontrolled, multiple time points, ITS design was used. The unit of analysis was the month in which women entered the 25th week of pregnancy. ITS models estimate the change in 'level' and 'trend' of the outcome of interest associated with the intervention. The change in level is the difference in intercepts between regression lines estimated from observations before and after the intervention. The change in trend is the difference in slopes. In the case of two 'interventions' (eg, introduction and withdrawal of the HiPG), two changes in level and trend are estimated.

Generalised least squares models were used allowing for autoregressive and moving average correlation structures as appropriate. These allow any effect of periodicity to be taken into account. First, associations between introduction and withdrawal of the HiPG and the outcomes of interest were assessed in the whole cohort, using separate models for each outcome. Final models were used to calculate estimated absolute and relative effects on each outcome of the introduction of the HiPG at 21 months after implementation (immediately prior to withdrawal), and 24 months after withdrawal, with 95% CIs.[26] Interaction terms were then used to determine whether the effects of the introduction or withdrawal of the HiPG varied by maternal age group, parity or IMD tertile.

Data preparation was conducted in Stata/SE V.14; data analysis in R V.3.3.1 and RStudio V.0.99.903. We used 95% CIs and a P value of <0.05 to indicate statistical significance throughout.

### RESULTS
#### Sample description

Of 39 571 women who delivered at the study hospital and were known to have reached the 25th week of gestation

**Table 2** Summary of interrupted time series models of the associations between the introduction and withdrawal of the Health in Pregnancy Grant and outcomes of interest, coefficients (95% CI)

| Model variable | Mean gestational age at booking (days) | Proportion booking by 10 weeks | Proportion booking by 18 weeks | Proportion booking by 25 weeks | Proportion of babies that were SGA |
|---|---|---|---|---|---|
| Time (months) | 0.02 (−0.01 to 0.05) | 0.0003* (−0.005 to 0.007) | **−0.0002 (−0.0003 to −0.00003)†** | −0.0001 (−0.0002 to 0.00005) | −0.00003 (−0.0003 to 0.0002) |
| Level change at introduction | **5.29 (2.47 to 8.11)** | −0.04 (−0.10 to 0.01) | **−0.03 (−0.04 to −0.02)** | **−0.03 (−0.05 to −0.02)** | 0.005 (−0.02 to 0.03) |
| Trend change at introduction | **−0.50 (−0.70 to −0.30)** | **0.005 (0.001 to 0.009)** | **0.003 (0.002 to 0.004)** | **0.003 (0.002 to 0.003)** | −0.001 (−0.002 to 0.001) |
| Level change at withdrawal | 1.37 (−1.63 to 4.37) | −0.03 (−0.08 to 0.02) | 0.001 (−0.01 to 0.02) | −0.01 (−0.02 to 0.004) | 0.01 (−0.01 to 0.03) |
| Trend change at withdrawal | **0.35 (0.13 to 0.57)** | −0.003 (−0.008 to 0.002) | **−0.002 (−0.003 to −0.001)** | **−0.002 (−0.003 to −0.001)** | 0.00 (−0.001 to 0.002) |

*Values are given to two decimal places or, for values <0.1, one significant figure.
†Bold indicates where 95% CIs do not cross 0.
SGA, small for gestational age.

between 1 January 2003 and 31 December 2013, full data were available for 34 589 (87.4%). Characteristics of those who did and did not meet the inclusion criteria and hence were included or excluded from the analysis are described in table 1. Most exclusions were due to missing information on birth weight. Typically, women included in the analyses were aged 25–34 years, of parity 1 or more, lived in the most deprived third of areas in England and booked by 10 weeks' gestation. Women excluded from the analyses tended to be younger, lived in more deprived areas and booked later in their pregnancies than women included. Similar differences between women included and excluded from the analyses were seen in each of the three study periods.

## Sample-wide changes in outcomes associated with introduction and withdrawal of the HiPG

Final models for each outcome are summarised in table 2 and plotted in figures 1–5. Introduction of the HiPG was associated with an immediate increase in mean gestational age at booking, and a decrease in the proportion booking by 18 and 25 weeks. That is, the immediate effect was for these outcomes to get clinically 'worse'. However, introduction of the HiPG was also associated with an improvement in the trend in mean gestational age at booking and proportion booking by 10, 18 and 25 weeks. That is, the longer term effect was a change in trend of these outcomes towards greater clinical improvement over time.

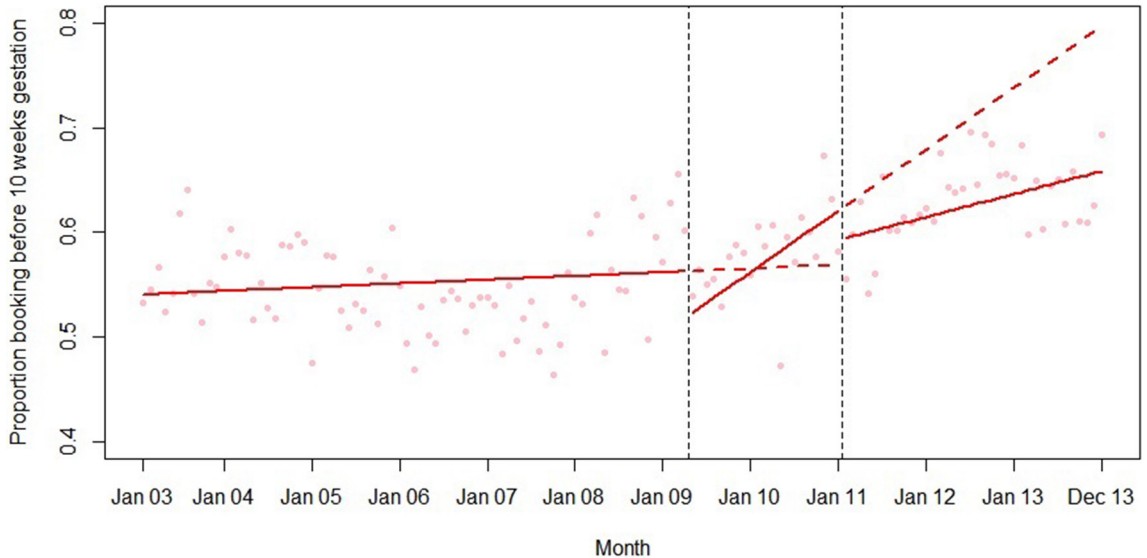

**Figure 2** Summary of interrupted time series model of the introduction and withdrawal of the Health in Pregnancy Grant on proportion booking before 10 weeks' gestation.

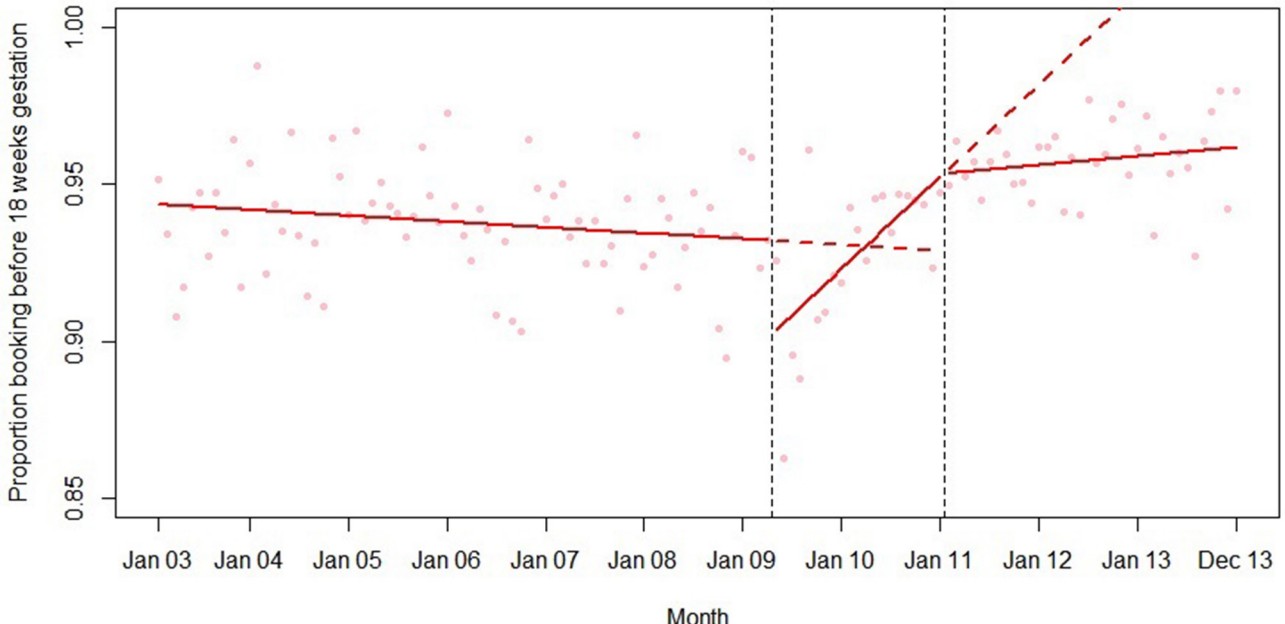

**Figure 3** Summary of interrupted time series model of the introduction and withdrawal of the Health in Pregnancy Grant on proportion booking before 18 weeks' gestation.

Withdrawal of the HiPG was not associated with any level changes in outcomes. However, it was associated with a change in trend in mean gestational age at booking and proportion booking by 18 and 25 weeks towards less clinical improvement over time. The introduction or withdrawal of the HiPG was not associated with any changes in the level or trend in the proportion of babies who were SGA.

Table 3 shows the absolute and relative impact of the introduction and withdrawal of the HiPG on each outcome at 21 months after introduction and 24 months

after withdrawal. By 21 months after introduction of the HiPG, compared with the counterfactual of what was predicted given trends prior to the introduction of the HiPG, there was a reduction in mean gestational age at booking of 4.8 days (95% CI 2.3 to 8.2), an increase in the proportion of women booking by 18 weeks of 2.2% (95% CI 1.2 to 3.9) and an increase in the proportion of women booking by 25 weeks of 1.9% (95% CI 0.6 to 3.5). Compared with the counterfactual of what was predicted to occur given trends when the HiPG was available, by 24 months after withdrawal, there was an increase in mean

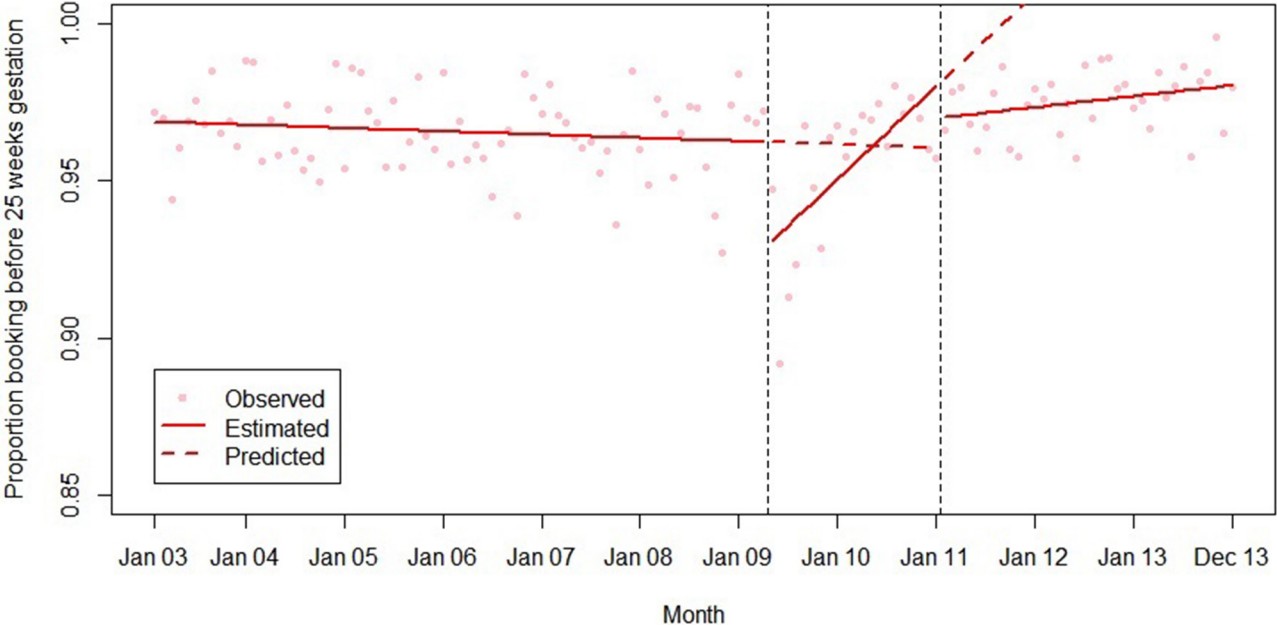

**Figure 4** Summary of interrupted time series model of the introduction and withdrawal of the Health in Pregnancy Grant on proportion booking before 25 weeks' gestation.

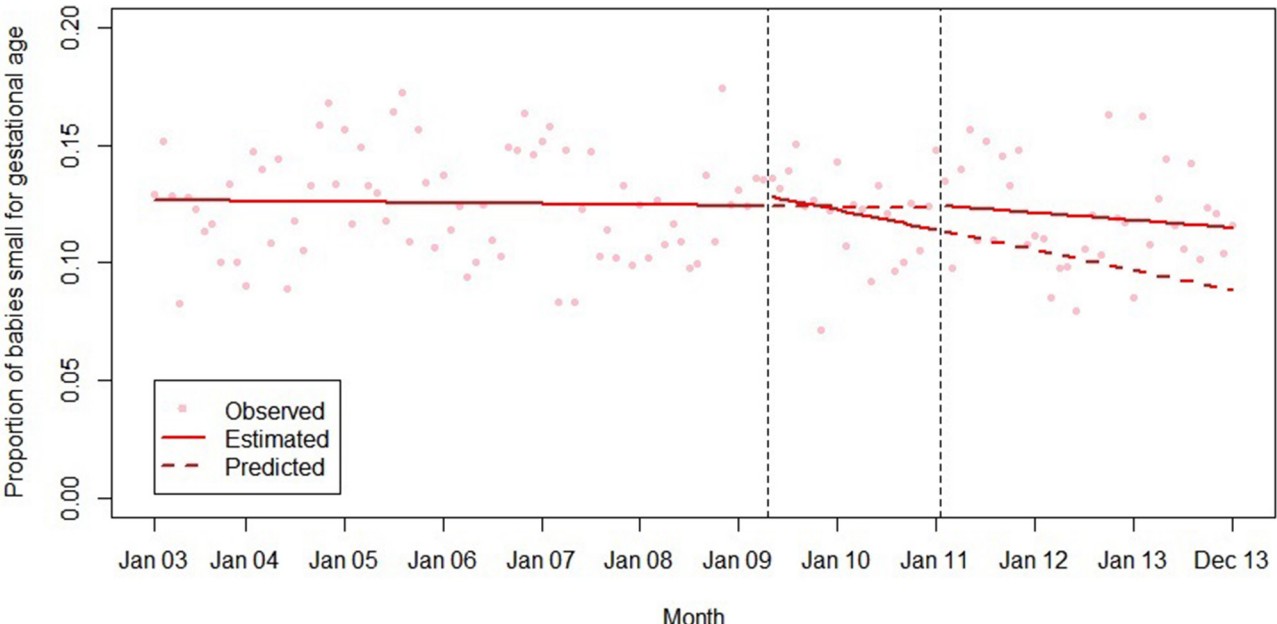

**Figure 5** Summary of interrupted time series model of the introduction and withdrawal of the Health in Pregnancy Grant on proportion of babies small for gestational age.

gestational age at booking of 14.0 days (95% CI 2.8 to 16.8), a decrease in the proportion of women booking by 18 weeks of 7.6% (95% CI 2.2 to 7.9) and a decrease in the proportion of women booking by 25 weeks of 8.3% (95% CI 3.1 to 8.6).

### Differential changes in outcomes associated with introduction and withdrawal of the HiPG across population subgroups

Models including interaction terms for maternal age, parity and IMD tertile are summarised in tables 4–6. There were no interactions with parity. The associations between introduction and withdrawal of the HiPG and trend in mean gestational age at booking varied by age group (figure 6), with greater changes in trend in older women.

The association between introduction of the HiPG and mean gestational age at booking and proportion booking by 18 and 25 weeks varied by IMD group (figures 7–9). The introduction of the HiPG was associated with a progressively larger level change (towards older gestational age at booking, and lower proportion booking by 18 or 25 weeks) as deprivation decreased.

### DISCUSSION
### Statement of principal findings

This is the first evaluation of the HiPG in England, the first evaluation of a financial incentive for attendance at antenatal care on incidence of SGA, and one of the largest pragmatic evaluations of a health-promoting financial incentive in a high-income country. Introduction of the HiPG was associated with immediate deteriorations in timing of booking, but longer term improvements over time. By 21 months after introduction of the HiPG (immediately prior to its withdrawal), mean gestational age at booking had decreased by 4.8 days compared with what would have been expected had it not been

**Table 3** Predicted effects (95% CI) of introduction and withdrawal of the Health in Pregnancy Grant at 21 months after introduction and 24 months after withdrawal

| Outcome | 21 months after introduction | | 24 months after withdrawal | |
| --- | --- | --- | --- | --- |
| | Absolute change | Relative change (%) | Absolute change | Relative change (%) |
| Mean gestational age at booking (days) | **–4.8 (–8.2 to –2.3)\*** | **–6.2 (–10.5 to –3.0)** | **14.0 (2.8 to 16.8)** | **25.2 (2.1 to 33.2)** |
| Proportion booking by 10 weeks | 0.06 (–0.02 to 12.5) | 10.3 (–4.2 to 22.6) | –0.14 (–0.24 to 0.03) | –17.4 (–26.8 to 1.3) |
| Proportion booking by 18 weeks | **0.02 (0.01 to 0.04)** | **2.2 (1.2 to 3.9)** | **–0.08 (–0.08 to –0.02)** | **–7.6 (–7.9 to –2.2)** |
| Proportion booking by 25 weeks | **0.02 (0.01 to 0.03)** | **1.9 (0.6 to 3.5)** | **–0.09 (–0.09 to –0.03)** | **–8.3 (–8.6 to –3.1)** |
| Proportion of babies small for gestational age | –0.01 (–0.03 to 0.01) | –8.1 (–25.9 to 10.8) | 0.03 (–0.03 to 0.08) | 29.8 (–57.4 to 104.9) |

*Bold indicates where 95% CIs do not cross 0.

**Table 4** Summary of interactions between parity and the introduction and withdrawal of the Health in Pregnancy Grant, model coefficients (95% CI)

| Model variable | Mean gestational age booking | Proportion booking by 10weeks | Proportion booking by 18weeks | Proportion booking by 25weeks | Proportion of babies that were SGA |
|---|---|---|---|---|---|
| Time | 0.03 (−0.01 to 0.08) | −0.0002 (−0.001 to 0.0004)* | −0.0002 (−0.0004 to 0.00001) | −0.0001 (−0.0003 to 0.00007) | −0.00002 (−0.0003 to 0.0003) |
| Parity >0 vs 0 | 0.09 (−2.54 to 2.72) | **−0.04 (−0.09 to −0.003)†** | 0.008 (−0.006 to 0.02) | 0.01 (−0.001 to 0.02) | −0.004 (−0.02 to 0.02) |
| Parity × time | −0.02 (−0.08 to 0.04) | 0.0005 (−0.0005 to 0.001) | 0.0002 (−0.002 to 0.002) | 0.00005 (−0.0002 to 0.0003) | −0.00002 (−0.0005 to 0.0004) |
| Level change at introduction | **5.57 (1.62 to 9.51)** | 0.01 (−0.05 to 0.07) | **−0.04 (−0.06 to −0.01)** | **−0.04 (−0.05 to −0.02)** | 0.01 (−0.02 to 0.04) |
| Trend change at introduction | **−0.50 (−0.78 to −0.21)** | 0.003 (−0.001 to 0.008) | **0.002 (0.001 to 0.004)** | **0.003 (0.001 to 0.004)** | −0.001 (−0.004 to 0.001) |
| Parity × level change at introduction | −0.61 (−6.19 to 4.97) | −0.04 (−0.13 to 0.04) | 0.007 (−0.02 to 0.04) | 0.007 (−0.02 to 0.03) | −0.01 (−0.06 to 0.03) |
| Parity × trend change at introduction | 0.01 (−0.39 to 0.41) | 0.0006 (−0.006 to 0.007) | −0.001 (−0.002 to 0.002) | −0.0002 (−0.002 to 0.002) | 0.002 (−0.002 to 0.005) |
| Level change at withdrawal | 1.03 (−3.16 to 5.22) | −0.02 (−0.08 to 0.05) | 0.005 (−0.02 to 0.03) | −0.01 (−0.03 to 0.001) | 0.02 (−0.04 to 0.02) |
| Trend change at withdrawal | 0.30 (−0.01 to 0.61) | −0.002 (−0.006 to 0.003) | **−0.002 (−0.003 to −0.00)** | −0.002 (−0.003 to 0.001) | 0.001 (−0.002 to 0.003) |
| Parity × level change at withdrawal | −0.13 (−5.80 to 6.06) | 0.006 (−0.09 to 0.10) | −0.003 (−0.04 to 0.03) | 0.001 (−0.03 to 0.03) | −0.03 (−0.07 to 0.02) |
| Parity × trend change at withdrawal | 0.08 (−0.36 to 0.52) | −0.001 (−0.008 to 0.006) | −0.001 (−0.003 to 0.002) | −0.001 (−0.002 to 0.002) | −0.001 (−0.004 to 0.003) |

*Values are given to two decimal places or, for values <0.1, one significant figure.
†Bold indicates where 95% CIs do not cross 0.
SGA, small for gestational age.

introduced. Withdrawal of the HiPG was not associated with any immediate changes in timing of booking, but it was associated with longer term deteriorations in timing of booking over time. By 24 months after withdrawal, mean gestational age at booking had increased by 14.0 days compared with what would have been expected had it not been withdrawn. We found no association between introduction or withdrawal of the HiPG and incidence of SGA. Trends in outcomes associated with the HiPG did not vary by parity. The positive association between introduction of the HiPG and gestational age at booking was greater in older women. The negative association between introduction of the HiPG and timing of booking was more pronounced in less deprived groups.

### Strengths and weaknesses of methods

The ITS approach is one of the strongest quasiexperimental research designs.[23 27] By including substantial data before and after interventions, we took account of underlying secular trends. By studying outcomes at the population level, rather than individual level, confounding by individual-level variables was avoided. Our large data set with 132 monthly data points substantially exceeds the minimum requirements for ITS.[27] By including autoregressive and moving-average functions, any biases introduced by the serial nature of the data (including seasonality and other periodicities) were accounted for. However, ITS designs are observational and we cannot categorically ascribe the changes documented to the HiPG. While we are not aware of any co-interventions likely to have influenced the outcomes concurrent with the HiPG, it is difficult to absolutely exclude these.

A major strength of our study is the use of SGA. Unlike a simple cut-off for low birth weight, SGA allows sex and gestational age differences in birth weight to be taken into account.

The data we used are likely to contain recording, reporting and transcription errors. Some of these may have introduced bias. 'Feasibility' limits were used for some variables and may have led to misclassification.

Cases included in the analytical cohort differed from those excluded. However, as differences between women included and excluded from the analyses were similar in all three study periods, this is unlikely to introduce bias and so we did not impute missing data. Differences between women included and excluded from the analyses may limit external validity, as may our use of data from only one hospital.

### Interpretation of findings

Our finding that the introduction of the HiPG was associated with an immediate deterioration in timing of booking may reflect an implementation phase, where the process for obtaining the HiPG was not yet fully understood. For instance, women may have thought that they were only entitled to the HiPG if they delayed attending until after the 25th week of pregnancy. In fact this was not the case—although women could not claim the grant

**Table 5** Summary of interactions between maternal age and the introduction and withdrawal of the Health in Pregnancy Grant, model coefficients (95% CI)

| | Mean gestational age booking | Proportion booking by 10 weeks | Proportion booking by 18 weeks | Proportion booking by 25 weeks | Proportion of babies that were SGA |
|---|---|---|---|---|---|
| Time | −0.01 (−0.06 to 0.05) | 0.00* (−0.001 to 0.002) | −0.0001 (−0.0004 to 0.0002) | −0.0001 (−0.0003 to 0.0001) | −0.0002 (−0.001 to 0.0002) |
| Age group | **−3.41 (−5.39 to −1.42)**† | **0.05 (0.002 to 0.11)** | **0.01 (0.004 to 0.02)** | 0.0002 (−0.006 to 0.007) | **−0.04 (−0.05 to −0.03)** |
| Age × time | 0.03 (−0.01 to 0.08) | −0.00 (−0.001 to 0.001) | −0.0001 (−0.0003 to 0.0001) | −0.0001 (−0.0002 to 0.0001) | 0.0002 (−0.0001 to 0.001) |
| Level change at introduction | 3.02 (−0.87 to 6.90) | −0.04 (−0.10 to 0.02) | −0.03 (−0.05 to −0.001) | **−0.02 (−0.04 to −0.004)** | −0.002 (−0.04 to 0.03) |
| Trend change at introduction | −0.25 (−0.53 to 0.04) | 0.002 (−0.003 to 0.008) | 0.002 (−0.0003 to 0.004) | **0.002 (0.00 to 0.003)** | 0.0003 (−0.002 to 0.003) |
| Age × level change at introduction | 2.15 (−0.86 to 5.16) | −0.01 (−0.06 to 0.04) | −0.003 (−0.02 to 0.02) | −0.005 (−0.02 to 0.001) | 0.005 (−0.02 to 0.03) |
| Age × trend change at introduction | −0.26 (−0.48 to −0.04) | 0.001 (−0.003 to 0.005) | 0.001 (−0.001 to 0.002) | 0.001 (−0.0002 to 0.002) | −0.001 (−0.003 to 0.001) |
| Level change at withdrawal | 0.57 (−3.45 to 4.59) | −0.04 (−0.10 to 0.02) | 0.01 (−0.02 to 0.04) | −0.006 (−0.03 to 0.02) | 0.007 (−0.03 to 0.05) |
| Trend change at withdrawal | 0.08 (−0.24 to 0.40) | −0.00 (−0.006 to 0.005) | −0.001 (−0.003 to 0.001) | −0.001 (−0.003 to 0.003) | −0.001 (−0.003 to 0.002) |
| Age × level change at withdrawal | 1.23 (−1.88 to 4.34) | 0.01 (−0.04 to 0.06) | −0.01 (−0.04 to 0.01) | −0.008 (−0.02 to 0.008) | 0.002 (−0.03 to 0.03) |
| Age × trend change at withdrawal | 0.27 (0.02 to 0.52) | −0.002 (−0.007 to 0.002) | −0.001 (−0.002 to 0.001) | −0.001 (−0.002 to 0.0004) | 0.001 (−0.001 to 0.003) |

*Values are given to two decimal places or, for values <0.1, one significant figure.
†Bold indicates where 95% CIs do not cross 0.
SGA, small for gestational age.

**Table 6** Summary of interactions between Index of Multiple Deprivation group and the introduction and withdrawal of the Health in Pregnancy Grant, model coefficients (95% CI)

| | Mean gestational age booking | Proportion booking by 10 weeks | Proportion booking by 18 weeks | Proportion booking by 25 weeks | Proportion of babies that were SGA |
|---|---|---|---|---|---|
| Time | **0.11 (0.03 to 0.19)*** | −0.001 (−0.003 to 0.001) | **−0.001 (−0.001 to −0.001)** | −0.0004 (−0.001 to 0.00003)† | 0.0003 (−0.0003 to 0.001) |
| Deprivation tertile | **4.53 (2.90 to 6.15)** | **−0.09 (−0.13 to −0.05)** | **−0.02 (−0.03 to −0.001)** | −0.005 (−0.01 to 0.003) | **0.04 (0.03 to 0.05)** |
| Deprivation × time | **−0.04 (−0.07 to −0.00)** | 0.0004 (−0.0004 to 0.001) | 0.0002 (−0.0001 to 0.0004) | 0.0001 (−0.0001 to 0.003) | −0.0001 (−0.0004 to 0.0001) |
| Level change at introduction | **17.18 (9.55 to 24.81)** | −0.06 (−0.17 to 0.06) | **−0.09 (−0.14 to −0.05)** | **−0.10 (−0.13 to −0.06)** | 0.01 (−0.05 to 0.07) |
| Trend change at introduction | **−1.04 (−1.59 to −0.49)** | 0.005 (−0.003 to 0.01) | **0.004 (0.001 to 0.007)** | **0.005 (0.002 to 0.008)** | −0.001 (−0.005 to 0.003) |
| Deprivation × level change at introduction | **−5.02 (−8.56 to −1.49)** | 0.007 (−0.05 to 0.06) | **0.03 (0.004 to 0.05)** | **0.03 (0.01 to 0.04)** | −0.001 (−0.03 to 0.03) |
| Deprivation × trend change at introduction | 0.22 (−0.03 to 0.47) | −0.0003 (−0.004 to 0.004) | −0.001 (−0.002 to 0.001) | −0.001 (−0.002 to −0.0002) | −0.00001 (−0.002 to 0.002) |
| Level change at withdrawal | 6.47 (−1.68 to 14.62) | −0.06 (−0.18 to 0.06) | −0.008 (−0.06 to 0.04) | −0.03 (−0.07 to 0.01) | 0.02 (−0.04 to 0.08) |
| Trend change at withdrawal | 0.59 (−0.005 to 1.18) | −0.002 (−0.01 to 0.01) | −0.002 (−0.006 to 0.001) | **−0.003 (−0.006 to −0.00002)** | −0.001 (−0.006 to 0.003) |
| Deprivation × level change at withdrawal | −2.10 (−5.88 to 1.67) | 0.01 (−0.04 to 0.07) | 0.004 (−0.02 to 0.03) | 0.008 (−0.01 to 0.03) | −0.003 (−0.03 to 0.03) |
| Deprivation × trend change at withdrawal | −0.10 (−0.37 to 0.18) | −0.00 (−0.005 to 0.005) | 0.00001 (−0.002 to 0.002) | 0.0004 (−0.001 to 0.002) | 0.001 (−0.001 to 0.003) |

*Bold indicates where 95% confidence intervals do not cross 0.
†values are given to two decimal places or, for values<0.1, one significant figure.
SGA, small for gestational age

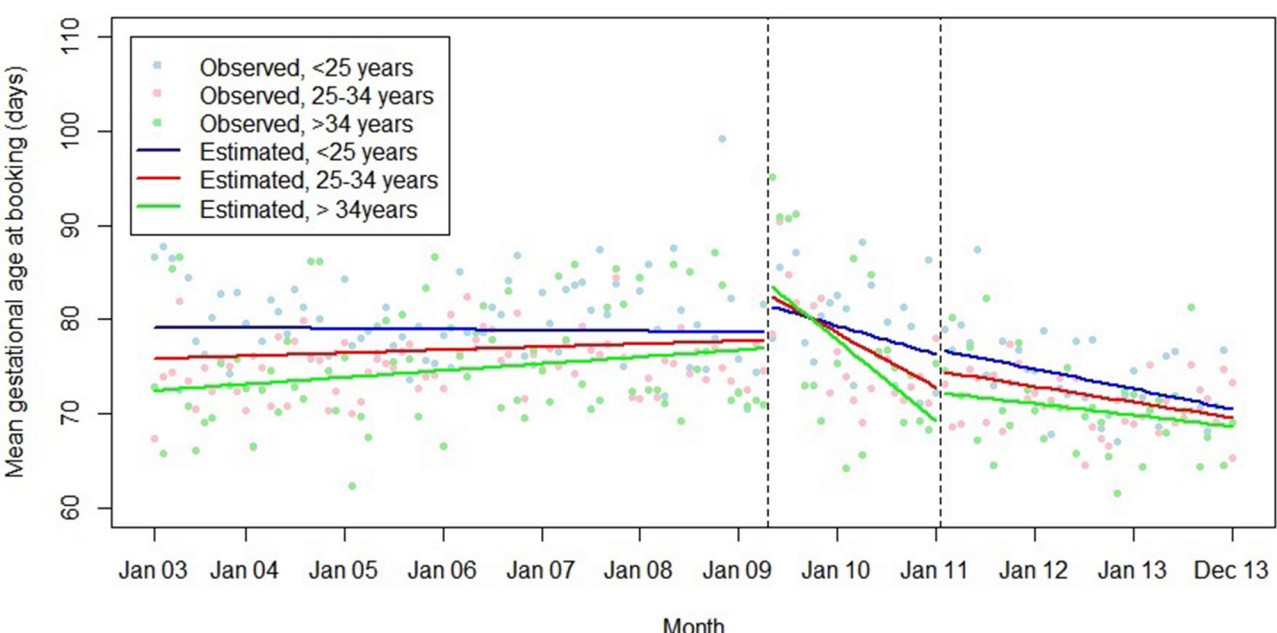

**Figure 6** Summary of interrupted time series model, interaction between maternal age group and the introduction and withdrawal of the Health in Pregnancy Grant on mean gestational age at booking (days).

until after the 25th week of pregnancy, whether they had first attended before this had no impact on their entitlement. While we could have conducted further analyses excluding an implementation period (eg, 4 months after introduction of the HiPG), this would have been post hoc justified.

The longer term associations of the introduction of the HiPG on markers of timing at booking are in line with the intention of the intervention—that women should attend for antenatal care earlier in their pregnancies. One previous study found no effect on timing of first

attendance of providing a voucher for a taxi journey to the antenatal clinic,[17] while a further evaluation of the HiPG found it was associated with a positive effect on the proportion of women booking by 25 weeks that disappeared after withdrawal.[19] The substantial difference in incentive value of the HiPG, compared with previous incentives, may explain these differences.

The finding that withdrawal of the HiPG was associated with deterioration of the benefits of its introduction on timing of booking is also not unexpected. On the whole, different women would have been pregnant when

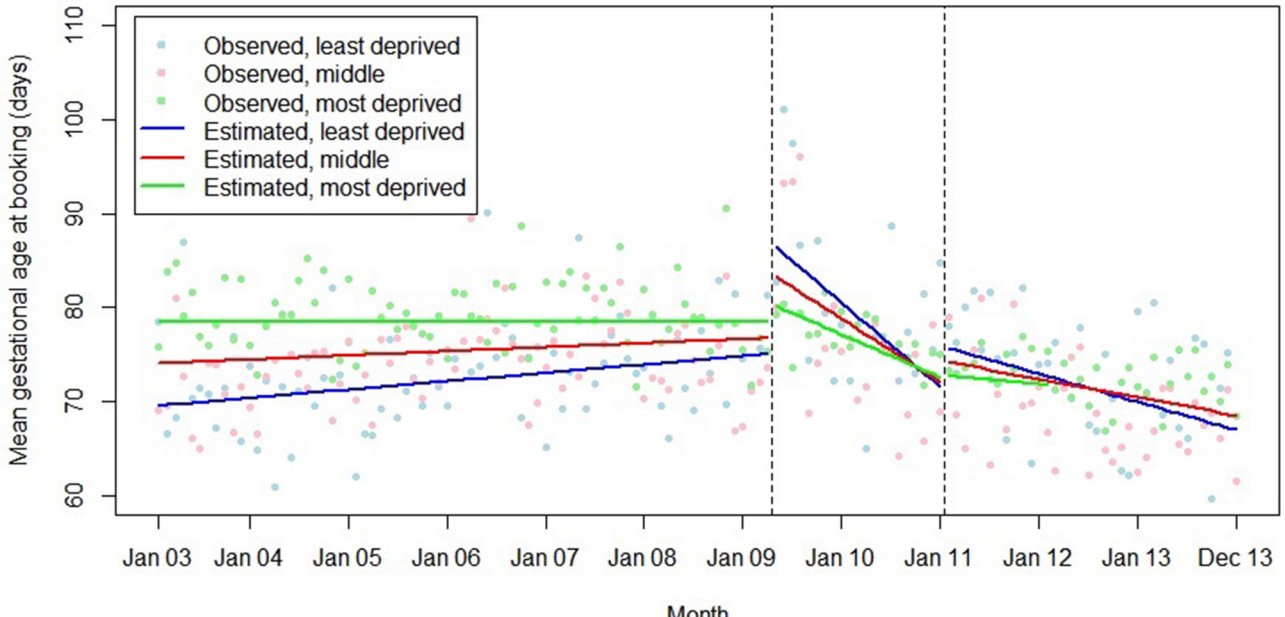

**Figure 7** Summary of interrupted time series model, interactions between Index of Multiple Deprivation group and the introduction and withdrawal of the Health in Pregnancy Grant on mean gestational age at booking (days).

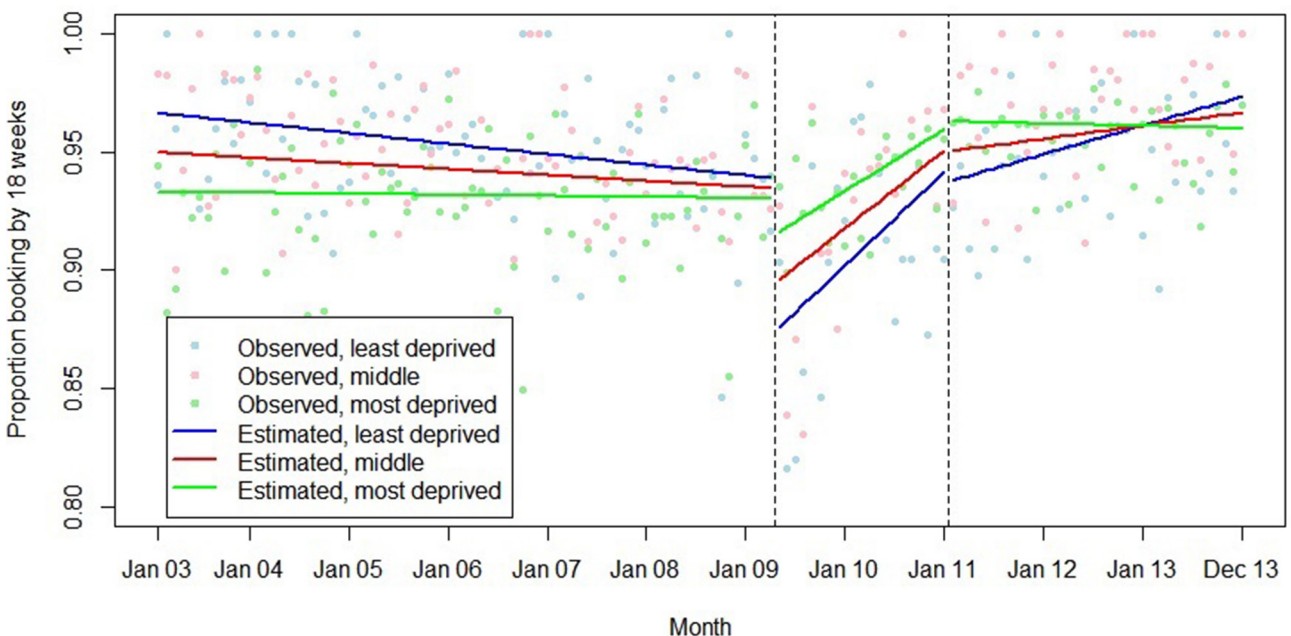

**Figure 8** Summary of interrupted time series model, interactions between Index of Multiple Deprivation group and the introduction and withdrawal of the Health in Pregnancy Grant on proportion booking before 18 weeks' gestation.

the HiPG was and was not available, meaning sustained effects would be highly unlikely.

Changes in timing of attendance for antenatal care associated with the introduction and withdrawal of the HiPG did not translate into differences in the proportion of SGA babies. This may be because the effect size (of 4.8 days at 21 months) was too small to impact on SGA. Two previous studies that examined the effect of incentives for antenatal care on incidence of low birth weight (rather than SGA) also reported no effect.[18 19]

Although the HiPG was only available from the 25th week of pregnancy, we found that its introduction was associated with changes in the proportion of women booking by both 10 and 18 weeks. This indicates that the impact of health-promoting financial incentives may not be as specific as previously thought.[28] The HiPG may have been associated with a larger effect on timing of booking if it had been contingent on booking earlier in pregnancy.

We did not find any evidence that the associations of the introduction or withdrawal of the HiPG with the outcomes

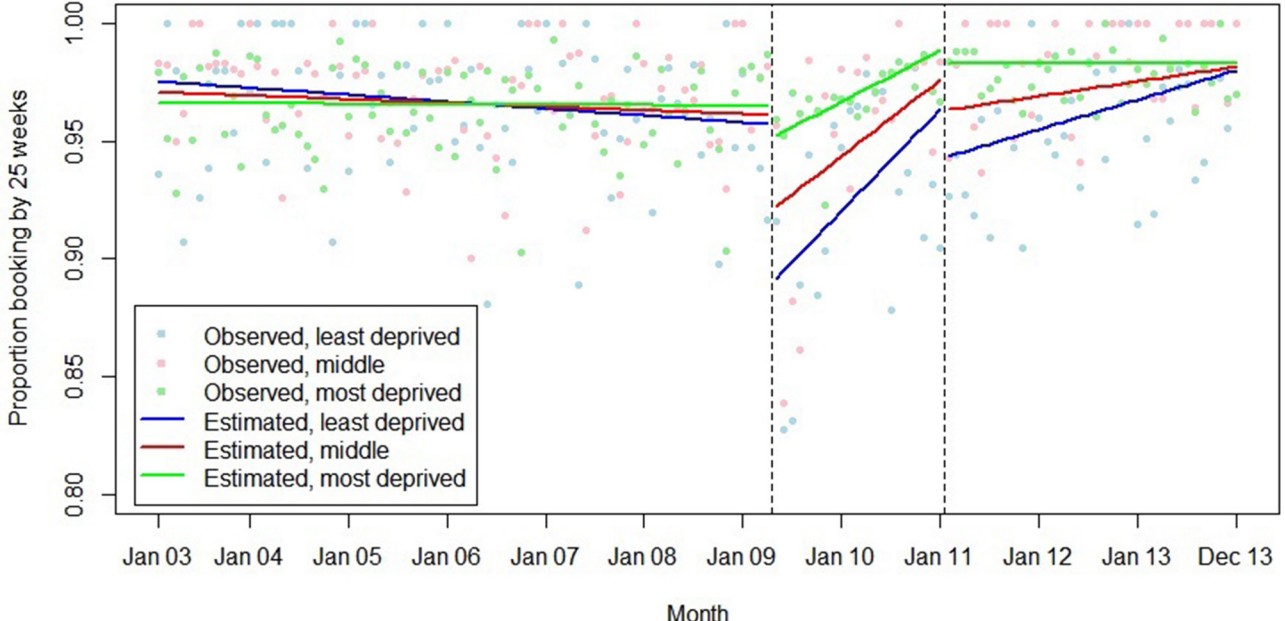

**Figure 9** Summary of interrupted time series model, interactions between Index of Multiple Deprivation group and the introduction and withdrawal of the Health in Pregnancy Grant on proportion booking before 25 weeks' gestation.

studied varied by parity. This suggests that prior experience of antenatal care did not diminish the impact of the HiPG. However, the association of the introduction of the HiPG with improvements in gestational age at booking over time was greater in older women. This suggests that age may be a determinant of responsiveness to financial incentives in this context, with older women being more responsive to the intervention. Our data are consistent with the suggestion that those with fewer resources are particularly responsive to incentives.[29] While introduction of the HiPG was associated with an immediate negative change in timing of first antenatal care, this was least pronounced in women living in the most deprived areas.

### Implications of findings for policy, practice and research

It is possible that larger incentives, contingent on attendance earlier than 25 weeks, may have greater impacts on timing of antenatal care and clinical outcomes than seen here. Future research could explore how effects on antenatal care attendance vary with incentive value and timing.

As we used routine data in a retrospective analysis conducted more than 2 years after withdrawal of the HiPG, we were unable to explore how women and other stakeholders responded to the HiPG. In particular, we do not know what women spent the HiPG on, how doctors and midwives discussed it with women, or how appropriate stakeholders thought it was. These factors may have influenced effectiveness and variations in effectiveness between subgroups.[30]

### CONCLUSIONS

Although the introduction of the HiPG was associated with an immediate clinical deterioration in timing of attendance for first antenatal care, it was also associated with a longer term trend towards improvement in timing. By 21 months after implementation, there was a decrease in almost 5 days in mean gestational age at booking compared with what would have been expected without implementation. Withdrawal of the HiPG was associated with deteriorations in timing of booking. By 24 months after withdrawal there was an increase in 14 days in mean gestational age at booking compared with what would have been expected without withdrawal. Neither the introduction nor withdrawal of the HiPG was associated with a change in proportion of babies who were SGA. There was no evidence that associations between introduction or withdrawal of the HiPG and outcomes varied by maternal parity. Introduction of the HiPG was associated with greater long-term benefits on timing at booking in older women, suggesting older women were most responsive to the intervention. The initial deterioration in timing of attendance for first antenatal care was least pronounced in those living in the most deprived circumstances, suggesting those living in the most deprived circumstances were most responsive to the intervention. Future research should explore the effects of incentives offered at different times in pregnancy and of differing values; and how stakeholders view such incentives.

**Contributors** JA conducted the literature searches, obtained the data, conducted the data analysis and led the writing. ZvdW, SR and JR contributed to study design, development of the analysis plan, interpretation of the data and critically reviewed previous versions of the final manuscript. JA acts as the guarantor.

**Funding** This work is produced under the terms of a Career Development Fellowship research training fellowship issued by the National Institute of Health Research (grant number CDF-2011-04-001) to JA. JA is supported by the Centre for Diet and Activity Research (CEDAR), a UKCRC Public Health Research Centre of Excellence. Funding from the British Heart Foundation, Cancer Research UK, Economic and Social Research Council, Medical Research Council, the National Institute for Health Research, and the Wellcome Trust, under the auspices of the UK Clinical Research Collaboration, is gratefully acknowledged (grant number MR/K023187/1).

**Disclaimer** The views expressed are those of the authors and not necessarily those of the NHS, the National Institute for Health Research or the Department of Health.

**Competing interests** None declared.

**Patient consent** Not required.

**Ethics approval** Ethics approval was granted by the East of England Norfolk NHS Research Ethics Committee (12/EE/0386). The routine hospital data used in this study were anonymised before transfer to the research team and the ethics committee determined that explicit patient consent was not required.

**Provenance and peer review** Not commissioned; externally peer reviewed.

**Data sharing statement** No additional data available.

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
