## [Reviewer comments · BMJ Open]

ARTICLE DETAILS

TITLE (PROVISIONAL)	Associations between introduction and withdrawal of a financial incentive and timing of attendance for antenatal care and incidence of small for gestational age: natural experimental evaluation using interrupted time series methods
AUTHORS	Adams, J; van der Waal, Zelda; Rushton, Steven; Rankin, Judith

VERSION 1 – REVIEW

REVIEWER	Christy Costanian York University. Toronto, Canada
REVIEW RETURNED	21-May-2017

GENERAL COMMENTS	Conclusion should be 1) made more clear and coherent , and 2) mention potential implications of the findings and future research needed.
--

REVIEWER	Clare Relton SchHARR, University of Sheffield. UK
REVIEW RETURNED	31-May-2017

GENERAL COMMENTS	This is an interesting well reported analysis of routine data on the HiPG and timing of attendance for ante natal care and incidence of small for gestational age. My comments are all minor There is a disparity between the primary outcome reported in the strengths and limitations section (p4, line 51) and the rest of the manuscript - abstract etc. The statement 'more deprived people may be more responsive to financial incentives' would benefit from a few sentences of further elucidation - is this supposition or is there evidence to support this statement (and what is the strength of this evidence) as deprivation is a thread running through the analysis - it would be good to have a firm basis at the start p10 line 193 "The positive effects of the introduction of the HiPG on trends in gestational age were greater in older women" I am not sure that the word effect is justified here? - suggest you use 'associated with'?
--

	(.....maybe older women are more confident about managing their pregnancies themselves ?) I suggest that the increase in gestational age at booking at the start - needs mentioning in the conclusions..... and could this increase be due to women delaying their booking in order to get the HiPG?
--	--

REVIEWER	Igor Locatelli University of Ljubljana, Faculty of Pharmacy Ljubljana, Slovenia
REVIEW RETURNED	07-Aug-2017

GENERAL COMMENTS	The manuscript "Effect of introduction and withdrawal of a financial incentive on timing of attendance for antenatal care and incidence of small for gestational age: natural experimental evaluation using interrupted time series methods" is very clearly written, is fluently readable and understandable. I have reviewed it with an emphasis on the statistical methods and analyses used. Regarding this I have some comments:  1. Study outcomes. Why other birth outcomes were not studied, e.g. proportion of preterm birth. This outcome is more easily obtained than SGA. It is also more important in terms of postnatal care of the newborn. 2. Monthly data points used. As your dataset is very large. You could use weekly data points and substantially improve the ITS model results (looking at the figures the data substantially deviates. It is a weak argument to say minimum requirements are covered; see line 98. Discuss other aspects why using monthly data points. 3. Univariate statistics for comparing included and excluded data. You have performed some statistics on raw data (i.e. without using statistical modeling). Please state in the Methods section the statistical tests used. They are only mentioned in the footnotes of the Table 1. (Chi2 and t tests) 4. Level of significance. You used $p < 0.01$ in univariate statistics, while a significant influence of parameters in Tables 2–6 was described by 95% CI (so type 1 error being 5%). Why this difference? You also did not explicitly state the value for significance level in the Methods. 5. A semi-independent sample. Statistical units (mothers) are not 100% independent. You mention that in the Lines 103-104 by stating that a "separate case" was applied for mothers with more than one pregnancy within the study period. However, it is not further explained, how these cases were modelled. In the results section these cases are not mentioned and even not discussed if this has any influence on study results. Please upgrade this view. 6. Why maternity age was categorized this way: <25, 25-34, or 35+ years. On what basis you decided for such categorization. Please add explanation to Line 115 7. Why z-scores below -3 or above 3 were excluded? It looks like you excluded the outliers from the statistical analysis with poor argumentation. Please add explanation to Line 124.
---

	8. Line 146. Difficult sentence to understand: was and was not available? 9. Line 241. Explain explicitly what the influence of age is, are older mother more or less responsive. 10. Line 260: You state: “Those living in more deprived circumstances showing the most positive initial response” while in line 243 you state “Whilst introduction of the HiPG was associated with an immediate negative change in timing of first antenatal care this was least pronounced in women living in the most deprived areas”. These two sentences are hard to be comparable. 11. Table 1. Variables booked by X days are categorical variables, so chi2 tests (mark 3) should be applied not t-tests (marked 4). 12. Table 1. This table represents the data. From descriptive statistics it could be seen that proportion booked by 10 weeks was constantly increasing (before HiPG = 55%; during HiPG = 58%; after HiPG = 63%). According to the results of ITS models different conclusions are written. Please add an explanation about this difference in Results and Discussion. The outcome Mean gestational age at booking, however, has a drop after HipG. This information should also be interpreted in view of ITS models results. 13. Table 2. It is very, very strange to write 0,00 (0.00 – 0.00) with CI intervals not crossing 0. Necessary, add the units behind time and try to represent the coefficients using larger time units (e.g. years). Or use scientific numbering. This is major issue. 14. Figures all of them. The figures would have more meaning if you add or shadow the some kind of predictive intervals onto them. 15. Figures all of them. Please make the X axis more explicit, where exactly lies e.g. Jun 07.
--	--

VERSION 1 – AUTHOR RESPONSE

In response to reviewer #1

Comment: “Conclusion should be 1) made more clear and coherent , and 2) mention potential implications of the findings and future research needed.”

Response: We have revised the conclusion section of both the abstract and main text to improve clarity and include potential implications for future research. Lines 41-46 and 287-304. Our revisions to the abstract have necessitated other changes to maintain it within the word limit.

In response to reviewer #2

Comment: “There is a disparity between the primary outcome reported in the strengths and limitations section (p4, line 51) and the rest of the manuscript - abstract etc.”

Response: We have now clarified in this section that proportion of babies that were small for gestational age was a secondary, not primary, outcome – as reported throughout the rest of the manuscript. Line 55.

Comment: "The statement 'more deprived people may be more responsive to financial incentives' would benefit from a few sentences of further elucidation - is this supposition or is there evidence to support this statement (and what is the strength of this evidence) as deprivation is a thread running through the analysis - it would be good to have a firm basis at the start"

Response: We have provided further justification for why deprivation may be interesting to study – because qualitative research often identifies that it may be more appropriate to target financial incentives at those living in more deprived circumstances. We have clarified that systematic review evidence suggests that fiscal interventions in general may be more effective in more deprived groups, but that differential response to health promotion financial incentives between population groups have not been systematically studied. Lines 83-87.

Comment: "p10 line 193 "The positive effects of the introduction of the HiPG on trends in gestational age were greater in older women" I am not sure that the word effect is justified here? - suggest you use 'associated with'? (.....maybe older women are more confident about managing their pregnancies themselves ?)"

Response: Thank you. We agree that 'associated with' is more appropriate and have changed this throughout the paragraph referred to. Lines 198-205.

Comment: "I suggest that the increase in gestational age at booking at the start - needs mentioning in the conclusions..... and could this increase be due to women delaying their booking in order to get the HiPG?"

Response: We have included this issue in the conclusions. Lines 287-288.

It is possible that women mistakenly believed that they were only entitled to the HiPG if they delayed attendance until after the 25th week of pregnancy. We have mentioned this possibility but clarified that this was not how the HiPG was designed. Lines 244-247.

In response to reviewer #3

"1. Study outcomes. Why other birth outcomes were not studied, e.g. proportion of preterm birth. This outcome is more easily obtained than SGA. It is also more important in terms of postnatal care of the newborn."

Response: The HiPG was designed to act as an "incentive to seek the recommended health advice at the appropriate time". Thus our outcome measures were primarily focused on when in pregnancy women sought antenatal care. We included proportion of babies who were small for gestational age (SGA) as a secondary outcome to assess any impacts of the intervention on infant 'health'. However, there is likely to be a long and complicated chain of causality between women receiving the HiPG and changes in infant health. Thus, whilst a number of alternative measures of infant 'health' may be available, we do not feel that these should be the focus of any evaluation. The HiPG could still be considered 'successful' in changing women's antenatal attendance behaviour, without any measurable effects on infant 'health'.

We have clarified our rationale for the outcomes used. Lines 117-118 and 123-126.

"2. Monthly data points used. As your dataset is very large. You could use weekly data points and substantially improve the ITS model results (looking at the figures the data substantially deviates. It is a weak argument to say minimum requirements are covered; see line 98. Discuss other aspects why using monthly data points."

Response: With 34,589 cases over 132 months we have a mean of 262 cases per month. However, if we instead divided the cases over the 572 weeks in the study period, we would have a mean of 60 cases per month. This would not have met the minimum suggested requirement of 100 cases per time point. As such, we chose to aggregate at the month, rather than week, level. We have clarified this in the methods section. Lines 107-109.

“3. Univariate statistics for comparing included and excluded data. You have performed some statistics on raw data (i.e. without using statistical modeling). Please state in the Methods section the statistical tests used. They are only mentioned in the footnotes of the Table 1. (Chi2 and t tests)”
We have added this information to the data analysis section of the methods as requested. Line 146.

“4. Level of significance. You used $p < 0.01$ in univariate statistics, while a significant influence of parameters in Tables 2–6 was described by 95% CI (so type 1 error being 5%). Why this difference? You also did not explicitly state the value for significance level in the Methods.”

Response: Thank you for noticing this. This was an error. We used $p < 0.05$ and 95% confidence intervals throughout. We have clarified this in Table 1 and stated it explicitly in the methods section. Lines 161-162.

“5. A semi-independent sample. Statistical units (mothers) are not 100% independent. You mention that in the Lines 103-104 by stating that a “separate case” was applied for mothers with more than one pregnancy within the study period. However, it is not further explained, how these cases were modelled. In the results section these cases are not mentioned and even not discussed if this has any influence on study results. Please upgrade this view.”

Response: As we did not have access to identifiable data on women (or their children), we were not able to determine on how many occasions individual women were included in the data set, or link such occurrences in order to take it into account in modelling. We have clarified this in the methods section. Lines 114-115.

“6. Why maternity age was categorized this way: <25, 25-34, or 35+ years. On what basis you decided for such categorization. Please add explanation to Line 115”

Response: We categorised age using mid-decade to mid-decade for the main grouping (25-34 years) as this convention is recommended by the International Journal of Epidemiology to increase comparability between studies (see https://academic.oup.com/ije/pages/Instructions_To_Authors). We did not further divide the <25 years group (e.g. into age 15-24 years and others) or the 35+ years group as (e.g. into age 35-44 years and others) as only 8 cases were aged less than 15 years, and only 27 were aged more than 44 years. We have provided additional information on this in the methods section as requested. Lines 133-136.

“7. Why z-scores below -3 or above 3 were excluded? It looks like you excluded the outliers from the statistical analysis with poor argumentation. Please add explanation to Line 124.”

Response: We excluded z-scores below -3 and above 3 as, in population data, these show significant deviation from the normal distribution and are expected to be caused by transcription errors (Bonellie et al (2008, ref 25 in manuscript). We have clarified this rationale in the methods section. Line 144.

“8. Line 146. Difficult sentence to understand: was and was not available?”

Response: We have revised this sentence to clarify that Table 1 describes the characteristics of cases that did and did not meet the inclusion criteria. Lines 167-168.

“9. Line 241. Explain explicitly what the influence of age is, are older mother more or less responsive.”

Response: We have clarified that the data are consistent with older women being more responsive to the intervention. Lines 272-273.

“10. Line 260: You state: “Those living in more deprived circumstances showing the most positive initial response” while in line 243 you state “Whilst introduction of the HiPG was associated with an immediate negative change in timing of first antenatal care this was least pronounced in women living in the most deprived areas”. These two sentences are hard to be comparable.”

Response: We agree, this is confusing. We have revised the later sentence to clarify that the negative initial effect was least pronounced in those living in the most deprived circumstances. Lines 299-301.

“11. Table 1. Variables booked by X days are categorical variables, so chi2 tests (mark 3) should be applied not t-tests (marked 4).”

Response: Thank you for noticing this error. We have corrected this throughout Table 1.

“12. Table 1. This table represents the data. From descriptive statistics it could be seen that proportion booked by 10 weeks was constantly increasing (before HiPG = 55%; during HiPG = 58%; after HiPG = 63%). According to the results of ITS models different conclusions are written. Please add an explanation about this difference in Results and Discussion. The outcome Mean gestational age at booking, however, has a drop after HipG. This information should also be interpreted in view of ITS models results.”

Response: The reviewer has identified an apparent discrepancy between the descriptive results and the fully adjusted results. The apparent discrepancy is due to adjustment. As we believe that the fully adjusted results are more ‘correct’, we do not feel it is appropriate to divert the reader’s attention away from these towards the less ‘correct’ unadjusted results only to then explain why the fully adjusted results are more ‘correct’. As such, we have not included additional text on this point.

“13. Table 2. It is very, very strange to write 0,00 (0.00 – 0.00) with CI intervals not crossing 0. Necessary, add the units behind time and try to represent the coefficients using larger time units (e.g. years). Or use scientific numbering. This is major issue.”

Response: Thank you for highlighting this. On reflection, we agree that this was a serious limitation. We have provided all figures to 2 decimal places, or 1 significant figure (for values less than 0.01) throughout all tables and updated footnotes to reflect this.

“14. Figures all of them. Please make the X axis more explicit, where exactly lies e.g. Jun 07.”
We have improved the readability of the x-axes for all figures by reducing the number of tick marks.

Response: We have not marked these changes to the figures as ‘track changes’.

VERSION 2 – REVIEW

REVIEWER	Clare Relton QMUL UK
REVIEW RETURNED	29-Nov-2017
GENERAL COMMENTS	Am happy with the revisions Just one comment - I do not see how this statement is derived from the reference that you cite. Please clarify or delete the statement. "There is some systematic review evidence that people living in Mmore deprived people circumstances may be more responsive to fiscal interventions in generalfinancial incentives..22 Other personal characteristics, such as age and previous experience of the behaviour incentivised, may also influence responsiveness".